# Nutrition and Immunity in Perinatal Hypoxic-Ischemic Injury

**DOI:** 10.3390/nu14132747

**Published:** 2022-07-01

**Authors:** Hema Gandecha, Avineet Kaur, Ranveer Sanghera, Joanna Preece, Thillagavathie Pillay

**Affiliations:** 1Department of Neonatology, University Hospitals Leicester NHS Trust, Leicester LE1 5WW, UK; hema.gandecha@uhl-tr.nhs.uk (H.G.); avineet.kaur@uhl-tr.nhs.uk (A.K.); ranveersingh.sanghera@nhs.net (R.S.); joanna.preece@uhl-tr.nhs.uk (J.P.); 2East Midlands Deanery, Health Education England, Leicester LE3 5DR, UK; 3Faculty of Science and Engineering, Research Institute for Healthcare Sciences, University of Wolverhampton, Wolverhampton WV1 1LY, UK; 4College of Life Sciences, University of Leicester, Leicester LE5 4PW, UK

**Keywords:** nutrition, immunity, newborn, IUGR (intrauterine growth restriction), hypoxia, ischaemia, nutritional immunology, nutrition immune axis

## Abstract

Perinatal hypoxia ischaemia (PHI), acute and chronic, may be associated with considerable adverse outcomes in the foetus and neonate. The molecular and cellular mechanisms of injury and repair associated with PHI in the perinate are not completely understood. Increasing evidence is mounting for the role of nutrients and bioactive food components in immune development, function and repair in PHI. In this review, we explore current concepts around the neonatal immune response to PHI with a specific emphasis on the impact of nutrition in the mother, foetus and neonate.

## 1. Introduction

Immune function in the foetus and newborn baby is continually evolving and influenced by a host of nutritional, cellular metabolic and immune–microbiome interactions. This review describes these influences and the impact of hypoxia and ischaemia on these interactions in the perinatal period. Hypoxia refers to a decrease in oxygen and ischaemia, due to a decrease in blood flow. If these happen in the perinatal period, i.e., in utero, or in a baby at birth or in the first few weeks after birth, it is referred to as perinatal hypoxic ischaemic injury (PHI).

In utero, this hypoxic ischaemic injury may be slowly progressive or chronic. It is usually the result of placental insufficiency (also known as placental dysfunction) in delivery of blood and therefore oxygen and nutrients to the foetus. This can result in foetal growth being suboptimal or restricted, commonly referred to as intrauterine growth restriction (IUGR). Common causes for placental insufficiency in utero include maternal pregnancy induced hypertension, diabetes, smoking and vascular malformations involving the placenta.

Both acute and chronic hypoxic-ischaemic injury have the potential to influence the inter-relationship between nutrition and immunity in the developing foetus and baby.

Hypoxic ischaemic injury in the perinatal period may also be acute. It may, for example, be due to placental inflammation (chorioamnionitis), placental abruption, uterine rupture, cord prolapse, or be associated with complicated deliveries such as cephalopelvic disproportion and with postnatal events such as hypoxia in respiratory failure from pneumonia or meconium aspiration [1].

## 2. Evolution of Immune Function in the Foetus and Perinatal Period

Immune function progressively remodels from foetal life in utero, after birth and throughout early childhood though remains limited in the perinatal period [2]. Development of immunity begins as early as 5 weeks in the embryo with the emergence of macrophages, followed by thymocytes, pre-β cells in liver and spleen, follicular dendritic cells and T and B cell development [3]. Concurrent tissue immune repertoires in the gastrointestinal tract, skin and respiratory tract (such as mucosal associated invariant T cells; innate lymphoid cells, langerhans cells and dendritic cells) also develop.

The foetus remains relatively immunologically tolerant as the parts of the immune system evolve [4,5]. Immune tolerance is important for three principal reasons: (a) to prevent rejection of the foetus, (b) to promote microbial colonisation after birth and (c) to prevent tissue damage from over-activation due to stress, injury and infection. Passive immunity from the mother is a critical component of immune protection both in utero and after birth (Figure 1, [6,7,8,9,10,11,12,13,14,15,16,17,18]). In these apparent quiescent phases, the immune system has the capacity to activate and respond to stressors that the foetus and neonate is exposed to in both specific and non-specific manners [6]. Examples of these can be seen in cytotoxic T cell immune responses to in utero infections such as with Human Immunodeficiency Virus [19], priming by in utero exposure to bacterial pathogens such as *Staphylococcus* and *Lactobacillus* [20], development of memory B cells in utero and in adaptations to physiologic and pathological hypoxia [21,22]. There is a fine balance between immune tolerance and the ability to mount a response to stressors such as infection and hypoxia and in this, nutrition plays an important multidirectional role (Figure 2, [18,23,24,25,26,27]).

## 3. Nutrition–Immunity Interdependence

### 3.1. Immunometabolism

Once an immune cell is activated, effector functions such as proliferation, cytokine production and chemotaxis are set in motion. There are at least six major metabolic pathways within immune cells that are important during immune activation [28]. These include glycolysis, the citric acid/Krebs/tricarboxylic acid (TCA) cycle, the pentose phosphate pathway, fatty acid oxidation, fatty acid synthesis and amino acid metabolism. The metabolic pathways used by different immune subsets differ; for example, under aerobic conditions, neutrophils utilise glucose and glycolysis as their source for ATP production, whereas T cells, B cells and monocytes principally use oxidative phosphorylation [29].

Any disruption of cellular metabolism can potentially have an impact on the effector function of the evolving immune system through changes in the availability of adequate substrate for high energy-requiring immuno-metabolic pathways. There is no translational neonatal research to consolidate this hypothesis but limited evidence from animal studies, suggests that the ketone body beta-hydroxybutyrate (βHB) can influence activation of the innate inflammatory response (NOD-like receptor protein 3 NLRP3) [30]. This in turn has the potential to influence release of pro-inflammatory cytokines. βHB is a fatty acid substrate which is transported to the liver and converted to Acetyl coenzyme A (coA). Acetyl coA then enters the TCA cycle and is used for ATP synthesis. Correlates for this in human studies do not yet exist but in theory in the context of perinatal events, this may promote activation of inflammation in the placenta and in foetal cellular tissue with subsequent hypoxia and its clinical consequences.

βHB may also have an antioxidative effect, which contributes to reduced oxidative stress damage through the oxidative pathway and suppresses inflammatory responses [30]. In animal studies an epigenetic effect in suckling rats has been demonstrated through its action as a histone deacetylase inhibitor [31]. This inhibitory action upregulates gene expression of brain derived neurotrophic factor thus promoting neuronal regeneration.

There are no robust explorative human foetal and neonatal studies on how nutritional deficiencies linked to the immuno-metabolic pathway may correlate with short- and long-term clinical manifestations of perinatal hypoxia-ischaemia (PHI) and on the potential ability of substrates to power up or quiesce immunological pathways. However, useful correlates from other mammalian foetal and neonatal research have been described [32].

### 3.2. Nutritional Immunology: Macro and Micronutrients and Immune Function

In an extensive review, Maggini explores the inter-relationship between immune function and nutrition across the ages [33]. In order to function, the immune system in humans is co-dependent on adequate macronutrient and micronutrient support from the earliest phases of immune development. Table 1 [34,35,36,37,38,39,40,41,42,43,44,45,46,47,48,49,50,51,52,53,54,55,56,57] outlines the most studied micronutrients and selected macronutrients that are essential for immunocompetence. In the perinatal period foetal nutrition is influenced by maternal nutrition and the transport and transfer of this across the placenta–foetal barrier. There are no extensive human studies of the dynamics of placenta–foetal transfer of macro and micronutrients or of the impact of their supplementation on foetal and neonatal immune function but, as outlined in Table 1, correlates of research in animal and cord blood studies exist.

### 3.3. The Nutrition–Microbiome–Immune Axis in the Foetus and Baby

Most human and animal studies have focused on a diet (nutrition)–microbiome–immune axis, or gut–immune axis [58,59]. In the foetus and newborn, the potential importance of the maternal microbiome is being understood, including the role of the microbiome in early human immunity [60,61,62]. Whilst existing studies do not show conclusive mechanisms of changes in the maternal microbiome or the impact of these changes, there is general agreement that there are changes in the maternal microbiome in the third trimester and these generally reflect a change in the microbial diversity [63]. The consequences of these changes are not yet fully understood.

The placenta has a natural non-pathogenic microbiome which may affect immune pathways during pregnancy and early foetal development [64]. The composition of these commensal communities reflects the composition of the microbiome of maternal diet and the oral cavity and may represent haematological spread. Neonates develop a complex microbiome within the first week of life and the composition of this fluctuates through the first few years. Exactly when the neonate is first exposed to bacteria is not known, but it is possible that first exposure comes from placental bacterial communities.

Creation of a healthy microbiome in the foetus and neonate that has a dynamic interaction with the host gastrointestinal metabolism establishes an adequate barrier and promotes absorption of nutrients, effective innate and adaptive immunity [65]. The presence of lactobacilli in the gut is known to alter the secretion of both interleukin-10 and interleukin-12 by macrophages which, in the presence of pathogens, results in an altered function of these cells and changed signalling to other components of the immune system. Antibody production by B cells can be influenced by the exposure to a wide diversity of appropriate gut microbiota and this may increase the efficacy of some aspects of the immune system.

Additional mechanisms, such as through epigenetic influences (nutri-epigenomics) assist the microbiome in contributing to healthy immune development [66]. Examples include transplacental passage of short chain fatty acids (SCFA) such as butyrate, that are produced through both diet and the maternal microbiome and which can influence epigenetics of the foetal immune repertoire. This potential effect of the maternal diet may also be reflected in the presence of these SCFA in breast milk, thus possibly providing further evidence of the link between the maternal microbiome and early immunity [66].

## 4. Immunological Changes in Perinatal Hypoxic Ischaemic Injury

In perinatal hypoxic ischaemic injury (PHI), decreased blood flow results in oxygen and nutrient depletion in the foetus and neonate. The brain is the most studied organ in relation to neonatal hypoxic ischaemic injuries (hypoxic ischaemic encephalopathy), together with studies of the impact of hypoxia and ischaemia on the neonatal gut [67,68]. Ischaemia triggers an inflammatory cascade and systemic, local and cellular immune mediated responses are activated. These cascades involve neutrophils, leucocytes, microglia, dendritic cells, macrophages, and lymphocytes, as well as the release of proinflammatory cytokines such as tumour necrosis factor alpha (TNF-α), interleukins (e.g., IL-1β and IL-6) and the antioxidant defence system [69,70]. In the brain, this results in the breakdown of the blood–brain barrier and infiltration of immune cells into the cerebral parenchyma leading to oedema and tissue damage. Immune cells release inducible nitric oxide synthase (iNOS), and this contributes to a negative effect of nitric oxide on cerebral ischaemia.

Following PHI, there is a brief recovery period if blood flow and oxygen are restored, followed by a secondary reperfusion injury 6 to 48 h after the initial injury that can last for days. This secondary injury appears to be related to oxidative stress damage, excitotoxicity and inflammation [71]. This may manifest in seizures, further cytotoxic oedema, excitotoxin release, impaired cerebral oxidative energy metabolism and eventually neuronal cell death.

The transcription factors hypoxia-induced factors (HIF), which enable tissues to adapt to hypoxia, have been studied in the context of PHI [72]. A potential relationship exists between HIF and amino acid metabolism in this situation.

### 4.1. Intrauterine Growth Restriction, Chronic Placental Insufficiency and Development of Immune Functions

In piglets, restriction of nutritional supply while inducing foetal hypoxia-ischaemia through obstruction of blood flow to the placenta results in foetal growth restriction with concomitant impaired immune development. This has been shown by altered plasma concentrations of interleukin 1β, immunoglobulin A, circulating lymphocytes, mRNA abundance of Toll-like receptor 9 and Toll-interacting protein in the ileum [73]. Similarly, in rat pups, T lymphocyte number and dual specificity protein phosphatase 1 (DUSP1), a CD4+ and CD8+ differentiating factor, were decreased in those born through artificially induced foetal hypoxia-ischaemia. These findings imply that foetal nutrition is critically important in optimal immune development in the baby.

There are no clinical human immunological studies in severely IUGR babies in whom an absolute hypoxic-ischaemic insult is known. However, animal correlates for altered immune function in the presence of foetal growth restriction (which may include hypoxia-ischaemia in placental inflammation) exist [24]. Further study is needed into the inter-relationship between maternal dietary deficiencies, microbiome variation and placental transfer or active transport of essential nutrients and microbiota in foetal and neonatal compromise.

Babies born growth restricted are at a higher risk for overall morbidity and mortality and this may in part be due to inter-related immunological factors. Interestingly, a detailed review has failed to show an independent association between IUGR and necrotising enterocolitis, unless in the presence of formula feed [74]. In a longitudinal study of Kenyan infants, there was a reduction in total lymphocyte counts at birth which persisted to 1 year in children with a birth weight below 2.5 kg [75]. Cell mediated immunity, as measured by response to tuberculin skin test, correlated significantly with birthweight, blood levels of iron and thiamin at birth, together with folate, pyridoxine and riboflavin levels at 6 and 12 months. This suggests an interrelationship between nutrition and immunity in IUGR babies.

### 4.2. Intrauterine Growth Restriction, Chronic Placental Insufficiency, Nutrition and Nutritional-Epigenetics

The impact of IUGR on the epigenetics of immune development and control has been studied. Key epigenetic mechanisms involved in gene expression such as histone modifications, DNA methylation, and non-coding RNAs may be affected by nutritional depletion [66]. Nutrients and bioactive food components that influence gene expression through DNA methylation at the time of immune differentiation include folic acid, choline, betaine, dietary fibres and carbohydrates. Vitamins and ethanol exert an influence through both DNA methylation and microRNA production and butyrate produced from gut microbiota can result in histone modifications. Thus, dietary factors, vitamins, ethanol and butyrate all have the ability to influence immune cellular development through the influence on gene expression of immune cells.

### 4.3. Intrauterine Growth Restriction and Autophagy

Autophagy is cellular autonomous immunity and is well defined in our knowledge of immunity. The intracellular nutrient environment and intracellular nutrient sensors mediate a link between autophagy and nutritional status. Autophagy increases in starvation and may in part be a protective mechanism although its role and contribution in chronic hypoxia and IUGR in humans is unknown. Recent studies in piglets demonstrate that maternal food restriction produced IUGR foetuses with intestinal injury associated with changes in cellular autonomous immunity via mTOR signalling pathways [76]. Similarly, in small intestines of IUGR neonatal piglets, autophagosomes, fewer epithelial goblet cells and lymphocytes, reduced levels of the cytokines TNF-α and IFN-γ and decreased gene expression of cytokines were seen, again suggesting an association between immune function and nutrition [77]. The correlates for this in the human foetus and neonate are unclear. Autophagy may be a protective response in trimming down resources to meet the available supply of nutrition and blood flow.

## 5. Acute Hypoxia-Ischaemia, Immunity, Nutrients and Trace Elements

In a well-grown baby with acute peri or postpartum PHI, the interplay of nutrition on immune function is not clearly defined [78]. Some evidence for the importance of appropriate nutritional support, directly and indirectly through creating the most appropriate microbiome for the baby is described below.

### 5.1. Glucose

The mechanisms for hypoxic ischaemic encephalopathy (HIE), one of the clinical consequences of PHI, include immune modulated cellular damage and inflammatory responses [70]. The brain is capable of utilising alternate energy sources such as ketone bodies, lactate and other fatty acids. In the presence of HIE and its associated impaired metabolic functions this ability to use alternate energy sources could be suboptimal as hypoglycaemia and hyperglycaemia are less well tolerated [32,79]. Short term hypoglycaemia and hyperglycaemia can have an effect on innate immune function [80,81,82]. Hyperglycaemia can aggravate injury to the thalamus, basal ganglia and brainstem more significantly than in hypoglycaemia [80,81]. Hyperglycaemia has also been shown to have an important association with developmental delay at 24 months of life, in preterm infants [83].

### 5.2. Amino Acids and Fatty Acids

Glucose and amino acids in early parenteral nutrition may also have an impact on immunity in the compromised neonate through augmentation of a hyperinflammatory response [84].

Experimental studies have demonstrated a role of omega 3 polyunsaturated fatty acids in counteracting neuroinflammation through limitation of immune cellular infiltration, release of proinflammatory mediators and excitatory glutamate and by restoration of mitochondrial function [25,85]. Post ischaemic administration of triglyceride emulsions containing docosahexaenoic acid has been shown to be neuroprotective in neonatal mice [25].

### 5.3. Micronutrients and Trace Elements

Term babies with HIE may have a lower selenium level independent of maternal levels [86]. Selenium supplementation ameliorates hypoxia-ischaemia induced neuronal death in vitro and in vivo. Studies on selenium pre-treatment in adult rats exposed to ischaemia showed a reduction in glutamate-induced reactive oxygen species production and preservation of mitochondrial membrane potential in the hippocampus with improvement in cellular respiration and complex motor and cognitive activities [87,88,89].

Iron is abundant in human cells, and microbes use this to flourish. This is closely linked to immune function through effects on the innate and adaptive immune system’s control of infection in the body. Low iron levels within the first few hours of birth decrease the risks for neonatal infection, and early supplemental iron increases risks for mortality and infection in the neonate [90].

Zinc is a key part of many cellular functions including integrity of the newborn skin, gastrointestinal and respiratory tract mucosa with supplementation in pregnancy in some groups having been shown to improve infant morbidity from diarrhoeal diseases in the first 6 months of life [44,91]. It is also believed to be required for normal growth of the foetus and through to puberty [92]. Zinc deficiency at important periods of brain development may significantly impact regulation of apoptosis and may play a role in non-specific and acquired immunity together with the function of key mediators of postnatal immune function [91,93].

## 6. Early Enteral Nutritional Support and Immunity in Perinatal Hypoxia-Ischaemia

### 6.1. Breast Milk and Colostrum

Colostrum and breast milk have immunoprotective and modulating factors including immunoglobulins, lactoferrin, oligosaccharides, growth factor peptides and cells. These all have the potential to influence innate and adaptive immune development and function in immature newborns [94,95]. In mothers infected with SARS-CoV-2 in the peripartum period, immune complexes in breastmilk have been shown to activate a mucosal immune response and develop the neonatal immune system beyond passive immunity [96].

Colostrum is rich in immune-regulating factors and may enhance immune defence locally and systemically if given into the oropharyngeal space (buccal colostrum) and enterally [97,98,99]. From studies in piglets, colostrum is known to have epigenetic programming properties and also reduces the risk for necrotising enterocolitis [100]. This effect is also seen in preterm babies, where the risks for necrotising enterocolitis (NEC) are highest in the preterm and especially in those with evidence of compromised growth due to placental insufficiency [101,102]. In IUGR preterm piglets, the incidence of NEC was higher in formula fed animals than in breast milk fed ones, and this trend is also seen in preterm human babies. This effect appears to be related to formula feed rather than presence of IUGR alone [103].

Both preterm and term babies with PHI are commonly managed with early parenteral nutrition and are often kept nil by mouth in the initial stabilisation period after birth. However, enteral nutrition with mothers’ breast milk and colostrum stimulates development of the gastrointestinal mucosa and enzyme systems. Delayed initiation of enteral feeds means that components in enteral feeds which promote mucosal growth and the evolution of an appropriate gut-microbiome axis are missing [104]. This could potentially affect immune development at gut trophic and systemic levels [104].

One of the key concerns in PHI is hypoxia ischaemia of the gastrointestinal tract, and a risk of developing necrotising enterocolitis (NEC). Recent studies have not shown a higher clinical correlation of NEC with PHI in those babies that required total body cooling, where feeds were started early or delayed [26]. In this population-based cohort UK study 31% of 6030 babies who had therapeutic hypothermia were fed during treatment [26]. Enteral feeding was associated with a lower pragmatic late onset sepsis rate, higher survival to discharge and a higher proportion of being breastfed at discharge when compared with the unfed group. The mechanisms of this are not clear but a nutrition–immune–microbiome interplay could be responsible via the passive immunity offered by maternal breast milk feeds and colostrum and the microbiome that establishes following early enteral breast milk and colostrum feeds [105].

### 6.2. Prebiotics and Probiotics

Probiotics have a protective role in immune mediation by modification of the cellular immune response. Probiotic therapy is used in the management of extreme preterm infants to optimise development of the neonatal gut microbiome and reduce the risk of necrotising enterocolitis in the neonatal period [106]. Studies in mice showed that *Lactobacillus reuteri* reduces the incidence and severity of experimentally induced NEC, potentially via the action of Toll-like receptor 2 (TLR2) which is an important component of innate immunity against bacterial pathogens [107]. Pre and probiotics have not been extensively studied in the context of PHI. However, early work supports the potential prebiotics and probiotics in relation to PHI, with evidence of a link between activation of pre-myelinating oligodendrocytes and the reduction in inflammation reducing secondary neuronal injury thus helping improve neurodevelopmental outcomes [108]. Human recombinant lactoferrin (rhLF) is a potential prebiotic. Data from studies where rhLF was injected during gestation or lactation into rat dams suggested a protective effect for cognitive development of the pups exposed to PHI [109]. The mechanism of this in relation to an immune association is unclear.

### 6.3. Neuroprotection

Therapeutic hypothermia aims to reduce secondary damage caused by PHI by reducing the brain metabolic demand and inhibiting the excito-oxidative cascade. It does not however, reduce susceptibility to the initial injury, or prevent brain injury from an initial hypoxic ischaemic event. Newer neuroprotective strategies for reversing or preventing the sequelae of neonatal HI are urgently required, along with research into the factors that drive brain plasticity and improved recovery. In adults with stroke, an advanced neuroprotective strategy includes neuroprotective diets which include bioactive components with antioxidant or anti-inflammatory properties [110]. Adequate nutrition during neonatal therapy from PHI is recognised as being potentially crucial in neuroprotection.

### 6.4. Prophylactic Nutraceuticals

Prophylactic nutraceuticals, specifically those dietary compounds with antioxidant, anti-inflammatory or anti-apoptotic properties are being studied now in the context of PHI [27]. This is a difficult area for research as acute HIE cannot be predicted, so most of the work in this area is being undertaken in rodent models. Pre-clinical studies in which HI was induced in rodent pups include utility of polyphenols, vitamins, omega-3 fatty acids, plant-derived compounds (such as capsaicin, tanshinones and sulforaphane) and endogenous compounds (such as carnitine, lactate and melatonin). Some of these are associated with decreased inflammation in areas such as the hippocampus, decreased apoptosis, decreased Toll-like receptor 4 (TLR4) and nuclear factor kB signalling. These are critical components of the innate immune response involved after a HI stress.

Immunomodulatory agents such as minocycline, IL-1 receptor antagonists, nuclear factor kB, Toll-like receptor agonists and stem cells [111], including those with anti-excitatory and anti- apoptotic function such as inter-alpha inhibitory proteins (IAIPs [112]), are being studied as adjuncts to cooling in treatment of HI but the clinical applicability and impact of such strategies has yet to be shown.

### 6.5. Placental Nutrient Sensing Maternal—Foetal Resource Allocation

The placenta has an important role to play in resource allocation between the mother and foetus. Hypoxia changes how the placenta supplies oxygen and nutrients. Amino acids play a crucial role in effector functions that are associated with pregnancy outcomes [113]. There are no major studies addressing the inter relationship between nutrient sensing in the placenta, hypoxia and interaction of the immune system.

## 7. Conclusions

From limited mammalian studies, a nutrition–immunity interrelationship in perinatal hypoxic ischaemic injury appears to exist with evidence of multifaceted interactions. Intrinsic nutritional components appear to drive alterations in the immune repertoire, interactions and function in utero and after birth together with the contribution of the maternal microbiome. Post-natal nutrient components can additionally modify the subsequent immune responses.

Further avenues of research are required to explore the contribution of nutrients to PHI and repair after injury. A better understanding of the role of the evolving microbiome–nutrition–immune relationship in newborn babies following PHI will facilitate the development of strategies to optimise support for effective immune function in PHI, to minimise injury in PHI and promote repair.

## Figures and Tables

**Figure 1 nutrients-14-02747-f001:**
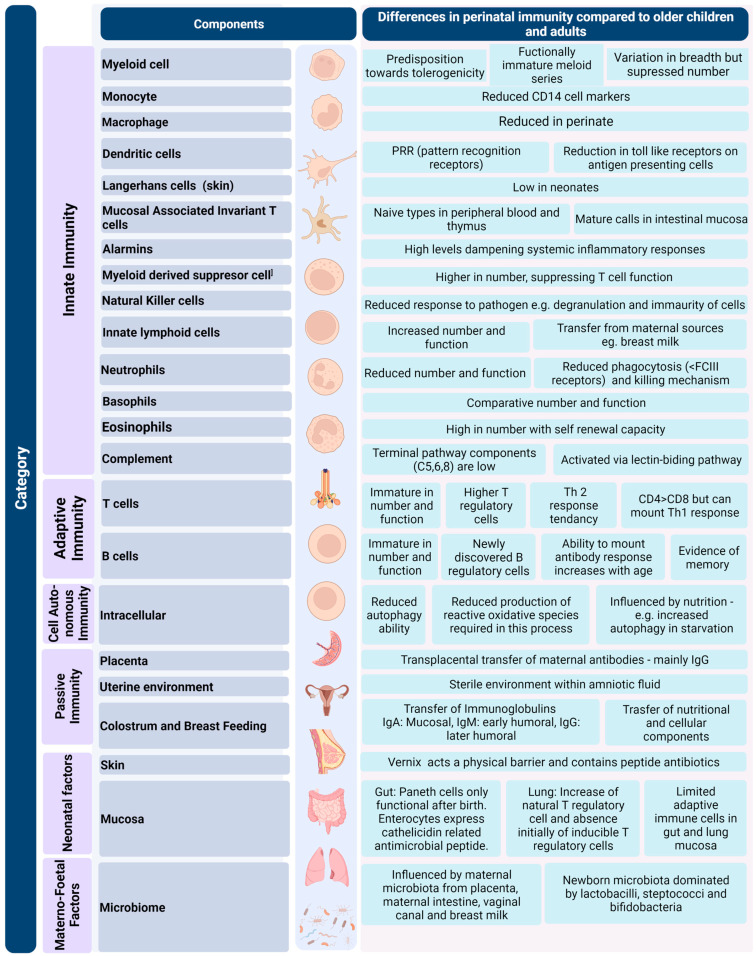
The perinatal immune repertoire: examples of differences in defence and immunity in the perinatal period compared to older children and adults. Image created using Biorender.com (accessed on 29 May 2022).

**Figure 2 nutrients-14-02747-f002:**
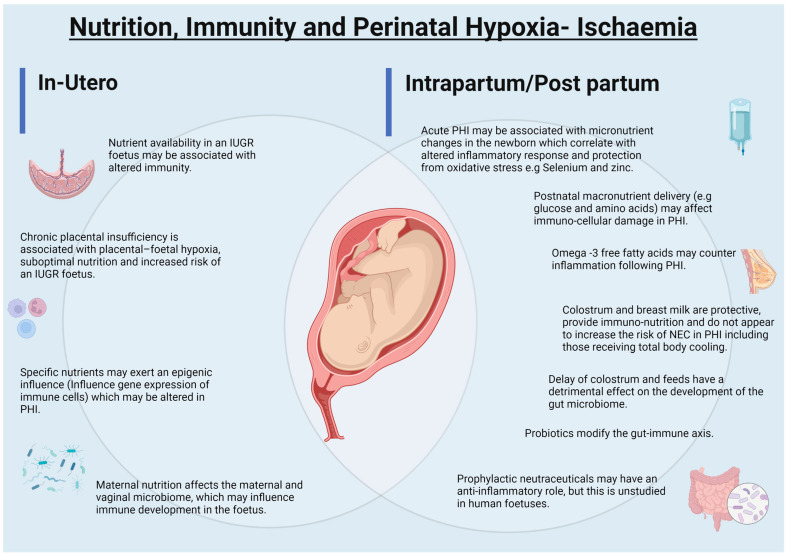
The nutrition–microbiome–immune axis in perinatal hypoxia-ischaemia (PHI): the interrelationship between perinatal nutrition, the maternal, placental and neonatal microbiome, and immunity (altered inflammatory response: injury, protection and repair) in the perinatal (foetal and neonatal) period is illustrated. Image created using Biorender.com (accessed on 29 May 2022).

**Table 1 nutrients-14-02747-t001:** Micronutrients and macronutrients: contribution to immune function, and potential impact of supplementation on perinatal immunity.

Nutritional Component	Contribution to Immune Function [23,33]		Potential Impact of Supplementation on Perinatal Immunity
MICRONUTRIENTS	Innate	Adaptive	
Vitamin A [34]	Structural and functional integrity of mucosal cells in innate barriers (e.g., skin, respiratory tract). Function of innate immune cells (e.g., natural killer (NK) cells, macrophages, neutrophils)	Important in T and B lymphocytes function. Involved in development and differentiation of Th1 and Th2 cells. Supports Th2 anti-inflammatory response	Adjuvant vitamin A in neonatal pneumonia increases IgM and IgG levels and shortens duration of infection Al-trans-retinoic acid supplementation in rat pups resulted in significantly higher levels of intestinal superoxide dismutase and glutathione peroxidase with reduced tissue tumour necrosis factor-α levels. These suggest a protective effect
Vitamin B [35,36,37,38]	Various B Vitamins impact function and activity of innate immune cells including dendritic cells (B6), NK cells (B6, B9/Folate, B12) and phagocytes (B2). B6 has a role in production of cytokine.	Important in synthesis and modulation of lymphocytes and activation of antibody production (B6, B9/Folate, B12) Role in supporting antibody response to antigens and Th1 response (B6, B9/Folate).	Maternal vitamin B12 supplementation may cause a slower decline in H1N1-IgG levels in neonates Antenatal supplementation with B9 in sheep was associated with increased levels of IgM and IgA in the offspring Maternal folic acid supplementation is associated with persistence of protective anti hepatitis B surface antigen five years after primary vaccination in the infant. Few other B vitamins have been studied in terms of their impact on immunity. Those that have shown little to no effect.
Vitamin C [39,40]	Antioxidant properties Promotes epithelial integrity Increases complement Promotes structure, function and movement of neutrophils, phagocytes and lymphocytes Important in NK cell activity and chemotaxis Role in apoptosis and clearance of neutrophils from infection site by macrophages	Increased antibody levels Increased lymphocyte differentiation and proliferation	Improved neutrophil chemotaxis in neonates with suspected sepsis Maternal vitamin C supplementation influenced cord blood mononuclear cell function by increasing cytokine (IFN-γ and IL-4) production, and decreasing IL-10 production
Vitamin D [41,42]	Promotes macrophage differentiation from monocytes Immune cell proliferation and cytokine production 1,25 dihydroxyvitamin D3 regulates defensins and cathelicidins (antimicrobial proteins that can directly kill pathogens)	Suppresses antibody production, inhibits T cell proliferation	Maternal Vit D correlates with leucocyte antigenic responses in breast feeding infants Newton 2022 High dose maternal vitamin D supplementation enhances proinflammatory cytokines response in cord blood. IL-17 A production increased (important in defence against respiratory pathogens)
Vitamin E [43]	Protects against free radical damage Enhances IL-2 and NK cell cytotoxic activity	Enhances T cell mediated function, promotes Th1 and suppresses Th2	Maternal peripartum supplement in calves improved phagocytic activity of neutrophils
Zinc [44]	Protects against free radical damage Modulates cytokines enhancing CD8+ proliferation Maintains physical immune barriers	Important in immune cell growth and differentiation Essential for T cell development and activation Supports Th1 response	Maternal supplementation improved IL-6 production and reduced number of episodes of diarrhoea in infants at 6 months of age
Iron [45]	Regulates cytokine production and function Supports killing of bacteria by neutrophilsImportant in the generation of free radicals	Supports differentiation and proliferation of T cells Component of enzymes essential for function of immune cells	Supplementation in neonates linked to increased Gram-negative infection. In vitro studies have shown overgrowth of pathogens that are implicated in neonatal sepsis in neonatal blood
Copper [46]	Free-radical scavenger Antimicrobial properties Important for IL-2 production and inflammatory response	Role in T cell proliferation, antibody production and cellular immunity	Perinatal supplementation in maternal cows increased antibody response and a reduction in respiratory infection in their calves
Selenium [47]	Essential for enzyme function (selenoproteins) counteracting free radicals Affects function of NK cells and leukocytes	Supports T cell proliferation Role in antibody mediated immunity	Systematic review identified a 12% reduction in incidence of late onset sepsis in very low birthweight neonates when supplemented postnatally with selenium
**MACRONUTRIENTS**	Innate	Adaptive	
Glucose/Oligosaccharide [48,49]	Metabolites are used as immune cell substrates. Type 2 innate lymphoid cells require glucose to proliferate Required for the effector functions of human NK cells, such as GLUT1, CD98 and CD71 Required for the activation of dendritic cells to express HLA-DR, CD80, CD86 and IFN-α	Role in class switch recombination in B cells Role in IFN-γ production from GAPDH.Helps express of Th2 cytokines (IL-4, IL-13) Supports proliferation of CD8+ T cells Role in activation of T reg cells	Innate: Increased cord blood cytokines (IL-6, IL-8 and TNF- α) when exposed to high glucose concentration post staphylococcal infection Inhibition of mTORC1 in murine NK cells prevents glycolysis required for granzyme b and IFN-γ production.Adaptive: required to drive proliferation and differentiation of CD4+ T cells in adult studies. Oligosaccharide diet may contribute to regulating Th1 cells in mice.
Amino acids [50,51]	Reduces TNF-α production by macrophages reducing the signalling to T-Lymphocytes High levels of adenosine increase cAMP, affecting neutrophil response and reduced expression of TNF-α. May also affect NK regulation. Glutamine: can affect eosinophil metabolic plasticity, required for T-cell function related to myeloid derived suppressor cells Arginine: required for T-cell function related to myeloid-derived suppressor cellsAlanine: a significant energy substrate for leucocytes Glycine: required in proliferation and antioxidative defence of leucocytesHistidine: required for the production of histamine required for macrophages and dendritic cell function	Glutamine: required for earliest stages of T-cell activation Asparagine: may modulate lymphocyte blastogenesis Aspartate is required lymphocyte proliferation Histidine: required for production of histamine required for T lymphocyte differentiation and function Lysine: required for proliferation of lymphocytes Tyrosine: the immediate precursor for catecholamine hormones, therefore important in the activation of T and B cells Serine: utilised for structural components and signalling in T and B cells.	Oral supplementation of glutamine enhances mucin synthesis in the small intestine of piglets. In rat pups and young piglets, dietary deprivation of glutamine has been associated with diminished intestinal integrity; supplementation improved growth, barrier function and protected against pathogen damage Arginine, glycine and histidine supplementation can improve immunological response. Lysine, tryptophan and tyrosine deficiency limits proliferation of lymphocytes and impairs response in chickens. Threonine improved outcome for immune responses in piglets challenged with E-Coli
Dietary Nucleotides [52,53]	Role in innate immunity Required for initial leukocytes stimulation	Required for initial lymphocyte activation	IUGR piglets have lower serum cytokine (IgA, IL-1β and IL-10), peripheral leucocyte levels and down regulation of innate immunity-related genes TOLLIP, TLR-9 and TLR-2. Dietary nucleotide supplementation improved peripheral leucocyte count, IgA and IL-1B and gene expression of TOLLIP, TLR-4 and TLR-9 in ileum A nucleotide free diet was associated with an increase in delayed cutaneous hypersensitivity, reduced NK cell and macrophage activity and spleen cell production of IL-2 in rodent studies
Glycoproteins [48,54]	Improved bactericidal properties of cells such as MDSC, e.g., with lactoferrin		Glutamine supplementation in low-birth-weight infants was associated with less translocated bacteria across the intestinal mucosa. This corresponded to a dampened immune response
Short chain Fatty acids (SCFA) (for example, Beta hydroxybutyrate) [35,55]	Ketone bodies: neutrophil effector function Fatty acid oxidation: expansion and cytokine production by Type 2 innate lymphoid cells	Differentiation of CD8+ t cells into T cytotoxic cells. Promotes CD4+ T cells IL-4 production in response to allergens. Promotes nitric oxide, IL-6 and TNF-α release Beta-hydroxybutyrate reduces B cell function	SCFA boosts the inflammatory process in murine studies. Maternal supplementation can reverse viral-induced islet inflammatory processes and therefore type 1 diabetes via modification of the microbiota in rat pups
Long chain Polyunsaturated Fatty Acids (LCPUFAs) [35,56,57]	LCPUFAs are used in cP450 pathway and produce Prostaglandins (PGs). PGs have multiple effects on dendritic cells, basophils, eosinophils, mast cells and macrophages LCPUFAs such as eicosapentaenoic acid (EPA) and docosahexaenoic acid (DHA) are utilised in pathways to produce specialised pro-resolving mediators DHA and arachidonic acid modulate Th1 and Th2 cell generation Can support tolerance and priming of the immune system (through suppressive IL-10 and transforming growth factor B)	Utilised in cP450 pathway and important in production of PGs. PGs have effects on Th1, Th2 cells, B cells, cytotoxic T cells, NK cells and on cytokine production such as interleukins and IFN gamma	Mice pups showed better responses to infections and vaccination in mothers supplemented with PUFAs during pregnancy LCPUFA formula fed infants were more likely to produce cytokines and lymphocyte populations similar to breast fed infants

## Data Availability

A full list of search items that were employed in the development of the concepts for the paper can be accessed by emailing the corresponding author. Publication licences for the figures are attached in the Appendix A.

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
