# Peer review of "Nutrition and Immunity in Perinatal Hypoxic-Ischemic Injury"

_nutrients, 2022, doi:10.3390/nu14132747_

Round 1

Reviewer 1 Report

The concept of this review is important but it is written with a very general manner and often lacks examples. It starts off without an introduction to set the stage. The reader has to guess what causes IUGR in the first place, and there are general statements, without specific examples given, throughout the text. The lack of line numbers has made this manuscript very hard to review. It is not possible to point out where there is a lack of clarity without citing page and describing contents, which is annoying. Please add line numbers to the revision if a revision is requested.

Why should interested readers have to email to the authors for a full list of search terms? Please provide directly as a supplementary table or a link to a publicly available archival repository.

Major:

The manuscript needs an introductory section laying out the scope of the review and providing generalities on the topic. This would be a good place to define IUGR and the circumstances that cause it,--when it occurs (risk factors), why (physiologically), how detected clinically? It is necessary to provide some context for readers of NUTRIENTS, who may not be perinatal specialists.

In the table, why are dietary nucleic acids placed under amino acids? What “other” amino acids are meant?

The text talks about response and repair. But it needs to be clearer to what is the baby responding? What repair is needed? Which organ systems? It was hard to know the scenarios being considered here that the foetus may be facing. There needs to be more explanation of what causes hypoxia in the first place.

What types of “immunomodulatory agents” are discussed here?

Minor points:

In utero shouldn’t be hyphenated

All-trans-retinoic acid (not altransretin...)

Some of the abbreviations are not spelled on first use; please provide abbreviations again in the table legends.

“Vitamin B”: It is important in the synthesis and modulation of what?  Is this all that the B vitamins do? Which B vitamins are being considered here? Folate is listed separately. This part seems skimpy.

Author Response

Professor T Pillay, on behalf of the authors

University Hospitals Leicester NHS Trust

University of Wolverhampton

tilly.pillay@uhl-tr.nhs.uk

T.Pillay@wlv.ac.uk

+447791936718

23 June 2022

To the Editor, 

Nutrients

Re: Peer Review Comments and Suggestions for Authors: Author Responses

Review article submission: Nutrition and immunity in perinatal hypoxic-ischemic injury. Gandecha Hema, Kaur Avineet, Sanghera Ranveer, Preece Joanna and Pillay Thillagavathie

Thank you for the peer review comments. These are addressed in the revised article as follows:

Reviewer 1

  1. a)  Reviewer 1 Comment: The concept of this review is important but it is written with a very general manner and often lacks examples. It starts off without an introduction to set the stage. The reader has to guess what causes IUGR in the first place, and there are general statements, without specific examples given, throughout the text. The lack of line numbers has made this manuscript very hard to review. It is not possible to point out where there is a lack of clarity without citing page and describing contents, which is annoying. Please add line numbers to the revision if a revision is requested.

Authors’  Response: Thank you. An Introduction, more references and line numbers are now included. These are detailed in the subsequent responses.

  1. b) Reviewer 1 Comment: Why should interested readers have to email to the authors for a full list of search terms? Please provide directly as a supplementary table or a link to a publicly available archival repository.

Authors’  Response: Thank you. A supplementary table has now been added to our submission.

Major:

  1. c) Reviewer 1 Comment: The manuscript needs an introductory section laying out the scope of the review and providing generalities on the topic. This would be a good place to define IUGR and the circumstances that cause it,--when it occurs (risk factors), why (physiologically), how detected clinically? It is necessary to provide some context for readers of NUTRIENTS, who may not be perinatal specialists.

 Authors’  Response:  Thank you. An Introduction that explains IUGR (the circumstances and why it occurs), and sets context for this work has been included. This is below and in rows 25-45 in the revised submission. The abstract has been modified to accommodate this introduction. The section numbering has also been modified due to the addition of an introduction.The first paragraph in section 3 (now section 4) has also been removed and incorporated in the new ‘Introduction’ 

‘Immune function in the foetus and newborn baby is continually evolving and influenced by a host of nutritional, cellular metabolic and immune-microbiome interactions. This review describes these influences, and the impact of hypoxia and ischaemia on these interactions in the perinatal period. Hypoxia refers to a decrease in oxygen and ischaemia, to a decrease in blood flow. If these happen in the perinatal period, i.e in utero, or in a baby at birth or in the first few weeks after birth, it is referred to as perinatal hypoxic ischaemic injury (PHI). 

In utero, this hypoxic ischaemic injury may be slowly progressive or chronic. It is usually the result of placental insufficiency (also known as placental dysfunction) in delivery of blood and therefore oxygen and nutrients to the foetus. This can result in foetal growth being suboptimal or restricted, commonly referred to as intrauterine growth restriction (IUGR).  Common causes for placental insufficiency in utero include maternal pregnancy induced hypertension, diabetes, smoking and vascular malformations involving the placenta.

Hypoxic ischaemic injury in the perinatal period may also be acute. It may for example, be due to placental inflammation (chorioamnionitis), placental abruption, uterine rupture, cord prolapse, be associated with complicated deliveries such as cephalopelvic disproportion and with postnatal events such as hypoxia in respiratory failure from pneumonia or meconium aspiration [1]. 

Both acute and chronic hypoxic-ischaemic injury have the potential to influence the inter-relationship between nutrition and immunity in the developing foetus and baby.’  

  1. d) Reviewer 1 Comment: In the table, why are dietary nucleic acids placed under amino acids? What “other” amino acids are meant?

Authors’ Response: Thank you for your insightful comment, Table 1 macronutrient section has now been revised as follows:   

  • Dietary nucleotides are no longer included under amino acids, but in a separate row
  • ‘Other amino acids’ has been changed to ‘Glycoproteins’
  1. e) Reviewer 1 comment: The text talks about response and repair. But it needs to be clearer to what is the baby responding? What repair is needed? Which organ systems? It was hard to know the scenarios being considered here that the foetus may be facing. There needs to be more explanation of what causes hypoxia in the first place.

Authors’  Response: Thank you for this comment . We realise that the way the sentences were structured could be confusing. We have therefore simplified it to reflect the current evidence. This described as follows in rows 113-116 on the revised document: 

‘In a well grown baby with acute peri or postpartum PHI, the interplay       of nutrition on immune function is not clearly defined [78]. Some evidence for the importance of appropriate nutritional support, directly and indirectly through creating the most appropriate microbiome for the baby is described below.’

The explanation of what causes hypoxia is now described in the introduction.

  1. f) Reviewer 1 Comment: What types of “immunomodulatory agents” are discussed here? 

Authors’ Response: : Thank you for this comment. This has been revised as follows: 

‘Immunomodulatory agents such as minocycline, IL-1 receptor antagonists, nuclear factor ?B, toll-like receptor agonists and stem cells [113], including those with with an anti-excitatory and anti- apoptotic function such as inter-alpha inhibitory proteins (IAIPs [112]),  are being studied as adjuncts to cooling in treatment of HI but the clinical applicability and impact of such strategies has yet to be shown. ‘

Minor points:

  1. g) Reviewer 1 Comments: In utero shouldn’t be hyphenated

All-trans-retinoic acid (not altransretin...)

Some of the abbreviations are not spelled on first use; please provide abbreviations again in the table legends.

 Authors’ Response: These have been addressed

  1. h) Reviewer 1 Comment: “Vitamin B”: It is important in the synthesis and modulation of what?  Is this all that the B vitamins do? Which B vitamins are being considered here? Folate is listed separately. This part seems skimpy.

Authors’ Response: Thank you for this comment. We have included folate as part of vitamin B and expanded this section to state which vitamin B components have shown to influence perinatal immunity.

Reviewer 2 Report

The Authors performed an extensive narrative review, with important concepts regarding the influences of nutrition on immune status of infants (from foetal to early life after birth), focusing on their role on perinatal hypoxic-ischemic injury.

I had appreciated a lot the manuscript, and the extensive biochemical-immunological explanations that the authors did. The work appears very well written, and the topic is studied very in deep.

Only some comments to improve the already good readability and quality of the study.

- There are some phrases without references (i.e. “there is general agreement that there are changes in the maternal microbiome in the third trimester and these generally reflect a reduction in the microbial diversity”; “Experimental studies have demonstrated a role of omega 3 polyunsaturated fatty acids in counteracting neuroinflammation through limitation of immune cellular infiltration, release of proinflammatory mediators and excitatory glutamate and by restoration of mitochondrial function”; “However, enteral nutrition with mothers breast milk and colostrum stimulates development of the gastrointestinal mucosa and enzyme systems. Delayed initiation of enteral feeds means that components in enteral feeds which promote mucosal growth and the evolution of an appropriate gut-microbiome axis are missing” et others), please provide;

- I want congratulated with you for the figures and table, they appear very claire and well done. However, as for table 1, I think that also figures (in the figure note or in the text after the mention) need references;

- “Hyperglycemia can aggravate injury to the thalamus, basal ganglia and brainstem more significantly than in hypoglycaemia”. It has been demonstrated also that, for preterm babies, hyperglycemia have an important role for neurodevelopmental delay, I recommend this information (https://doi.org/10.3390/nu13061930);

- “4.3 Micronutrients and trace elements”: It has been studied the role of zinc in early life for immunity, NEC, morbidity and neurodevelopment. Why this element has not been mentioned in this section? Please, provide;

- “These all have the potential to influence innate and adaptive immune development and function in immature newborns”. It has been recently suggested how maternal protection goes beyond passive immunity, with immune complexes in breastmilk stimulating the active development of the neonatal immune system, in SARS-CoV-2 infection. Could be interestingly add this concept.

- Please note that "NEC" is not abbreviated at the first mention

Congratulations for the very good work!

Author Response

Professor T Pillay, on behalf of the authors

University Hospitals Leicester NHS Trust

University of Wolverhampton

tilly.pillay@uhl-tr.nhs.uk

T.Pillay@wlv.ac.uk

+447791936718

23 June 2022

To the Editor, 

Nutrients

Re: Peer Review Comments and Suggestions for Authors: Author Responses

Review article submission: Nutrition and immunity in perinatal hypoxic-ischemic injury. Gandecha Hema, Kaur Avineet, Sanghera Ranveer, Preece Joanna and Pillay Thillagavathie

Thank you for the peer review comments. These are addressed in the revised article as follows:

Reviewer 2

  1. a) Reviewer 2 Comment: The Authors performed an extensive narrative review, with important concepts regarding the influences of nutrition on immune status of infants (from foetal to early life after birth), focusing on their role on perinatal hypoxic-ischemic injury.

I had appreciated a lot the manuscript, and the extensive biochemical-immunological explanations that the authors did. The work appears very well written, and the topic is studied very in deep.

Authors’ Response: Thank you for your kind comments.

  1. b) Reviewer 2 Comment: Only some comments to improve the already good readability and quality of the study.

- There are some phrases without references, i.e :

“there is general agreement that there are changes in the maternal microbiome in the third trimester and these generally reflect a reduction in the microbial diversity”;

 Authors’ Response: A reference has been added, reference number [63] - (Nuriel-Ohayon M, Neuman H, Koren O. Microbial Changes during Pregnancy, Birth, and Infancy. Front Microbiol. 2016;7:1031. Published 2016 Jul 14. doi:10.3389/fmicb.2016.01031) 

  1. c) Reviewer 2 Comment: “Experimental studies have demonstrated a role of omega 3 polyunsaturated fatty acids in counteracting neuroinflammation through limitation of immune cellular infiltration, release of proinflammatory mediators and excitatory glutamate and by restoration of mitochondrial function”; 

 Authors’ Response:  References have been added, reference numbers [25,85]. 

Williams JJ, Mayurasakorn K, Vannucci SJ, et al. N-3 fatty acid rich triglyceride emulsions are neuroprotective after cerebral hypoxic-ischemic injury in neonatal mice. PLoS One. 2013;8(2):e56233. doi:10.1371/journal.pone.0056233

Joffre C, Rey C, Layé S. N-3 Polyunsaturated Fatty Acids and the Resolution of Neuroinflammation. Front Pharmacol. 2019;10:1022. Published 2019 Sep 13. doi:10.3389/fphar.2019.01022

  1. d) Reviewer 2 Comment: “However, enteral nutrition with mothers breast milk and colostrum stimulates development of the gastrointestinal mucosa and enzyme systems. Delayed initiation of enteral feeds means that components in enteral feeds which promote mucosal growth and the evolution of an appropriate gut-microbiome axis are missing” et others), please provide;

Authors’  Response: A reference has been added, reference number [104]

Morgan, W.; Yardley, J.; Luk, G.; Niemiec, P.; Dudgeon, D. Total Parenteral Nutrition and Intestinal Development: A Neonatal Model. Journal of Pediatric Surgery 1987, 22, 541–545, doi:10.1016/S0022-3468(87)80217-8.

  1. e) Reviewer 2 Comment:  I want to congratulate you for the figures and table, they appear very clear and well done. However, as for table 1, I think that also figures (in the figure note or in the text after the mention) need references;

 Authors’ Response: Thank you for your comments. Both figures are now referenced.

  1. f) Reviewer 2 Comment: “Hyperglycemia can aggravate injury to the thalamus, basal ganglia and brainstem more significantly than in hypoglycaemia”. It has been demonstrated also that, for preterm babies, hyperglycemia have an important role for neurodevelopmental delay, I recommend this information (https://doi.org/10.3390/nu13061930); 

 Authors’ Response: Thank you for your useful comment and this reference, these have been added to the text, under section 5.1:

‘Hyperglycaemia has also been shown to have an important association with developmental delay at 24 months of life, in preterm infants [83].’ 

  1. g) Reviewer 2 Comment: “4.3 Micronutrients and trace elements”: It has been studied the role of zinc in early life for immunity, NEC, morbidity and neurodevelopment. Why this element has not been mentioned in this section? 

 Authors’ Response: Thank you for noting this. Although we had mentioned zinc in table 1, we have now expanded this in the text as recommended in section 5.3.

‘Zinc is a key part of many cellular functions including integrity of the newborn skin, gastrointestinal and respiratory tract mucosa with supplementation in pregnancy in some groups having been shown to improve infant morbidity from diarrhoeal diseases in the first 6 months of life [44,91]. It is also believed to be required for normal growth of the foetus and through to puberty [92]. Zinc deficiency at important periods of brain development may significantly impact regulation of apoptosis and may play a role in non-specific and acquired immunity together with the function of key mediators of postnatal immune function [91,93].’    

  1. h) Reviewer 2 Comment: Please, provide; - “These all have the potential to influence innate and adaptive immune development and function in immature newborns”. It has been recently suggested how maternal protection goes beyond passive immunity, with immune complexes in breastmilk stimulating the active development of the neonatal immune system, in SARS-CoV-2 infection. Could be interestingly add this concept. 

 Authors’ Response: Thank you very much for this insightful comment. We have now included work around SARS-CoV in section 6.1 as below:  

‘In mothers infected with SARS-Cov-2 in the peripartum period, immune complexes in breastmilk have been shown to activate mucosal immune response and develop the neonatal immune system beyond passive immunity [96].’

Reference: (Conti MG, Terreri S, Piano Mortari E, et al. Immune Response of Neonates Born to Mothers Infected With SARS-CoV-2. JAMA Netw Open. 2021;4(11):e2132563. Published 2021 Nov 1. doi:10.1001/jamanetworkopen.2021.32563) 

  1. i) Reviewer 2 Comment: Please note that "NEC" is not abbreviated at the first mention.

Authors’ Response: Thank you for this, we have amended this in the text. 

  1. j) Reviewer 2 Comment: Congratulations for the very good work!

Authors’ Response: Thank you very much

Sincerely

T Pillay

T Pillay, on behalf of the authors

Round 2

Reviewer 1 Report

This is much improved and now better addresses itself to a general audience. The topic is important and not often covered in nutritional sciences journals.

One point remains: the Data Availability Statement still says that a full list of search items that were employed in the development of the concepts for the paper can be accessed through emailing the corresponding author" but the response to the review says that this is now in a supplementary table 1. Please include the table and revise this statement.